# Natural Products Targeting Amyloid Beta in Alzheimer’s Disease

**DOI:** 10.3390/ijms22052341

**Published:** 2021-02-26

**Authors:** Joo-Hee Lee, Na-Hyun Ahn, Su-Bin Choi, Youngeun Kwon, Seung-Hoon Yang

**Affiliations:** Department of Medical Biotechnology, College of Life Science and Biotechnology, Dongguk University, Seoul 04620, Korea; 2018111727@dgu.ac.kr (J.-H.L.); skgus9506@dgu.ac.kr (N.-H.A.); dahee1546@dgu.ac.kr (S.-B.C.); ykwon@dgu.ac.kr (Y.K.)

**Keywords:** Alzheimer’s disease, natural products, amyloid beta, secretase-dependent, structure-dependent

## Abstract

Alzheimer’s disease (AD) is a neurodegenerative disease characterized by severe brain damage and dementia. There are currently few therapeutics to treat this disease, and they can only temporarily alleviate some of the symptoms. The pathogenesis of AD is mainly preceded by accumulation of abnormal amyloid beta (Aβ) aggregates, which are toxic to neurons. Therefore, modulation of the formation of these abnormal aggregates is strongly suggested as the most effective approach to treat AD. In particular, numerous studies on natural products associated with AD, aiming to downregulate Aβ peptides and suppress the formation of abnormal Aβ aggregates, thus reducing neural cell death, are being conducted. Generation of Aβ peptides can be prevented by targeting the secretases involved in Aβ-peptide formation (secretase-dependent). Additionally, blocking the intra- and intermolecular interactions of Aβ peptides can induce conformational changes in abnormal Aβ aggregates, whereby the toxicity can be ameliorated (structure-dependent). In this review, AD-associated natural products which can reduce the accumulation of Aβ peptides via secretase- or structure-dependent pathways, and the current clinical trial states of these products are discussed.

## 1. Introduction

Alzheimer’s disease (AD) is the most common type of dementia and neurodegenerative disorder, characterized by severe loss of neural cells, eventually leading to the loss of memory. The exact cause of AD is still unknown, but abnormalities related to amyloid beta (Aβ) peptides, such as misfolded Aβ aggregation, are observed in the brains of AD patients. During the progression of AD, Aβ monomers aggregate into oligomers and fibrils, which are toxic to neural cells, thus causing brain damage [1]. These aggregates further accumulate in the brains of AD patients to form Aβ plaques. The detailed mechanisms underlying the accumulation of these plaques have been investigated by many research groups [2,3], and it is certain that Aβ deposition is generally observed in the brains of AD patients before the symptoms, such as loss of memory, fully manifest, except for the patients who are non-demented with Alzheimer’s neuropathology (NDAN) [4,5].

Aβ peptides are first produced through cleavage of amyloid precursor proteins (APPs). When APPs are synthesized in the brain, they then translocate to the plasma membrane and undergo specific endoproteolytic cleavages by α-, β-, and γ- secretases. Generally, APPs are processed via either of two pathways, amyloidogenic and non-amyloidogenic [2]. In the non-amyloidogenic pathway, APPs are cleaved by α-secretase, releasing the soluble N-terminal fragment (sAPPα) into the extracellular space and leaving the membrane-tethered C terminal fragment (CTFα, also called C83). CTFα undergoes intramembrane cleavage by γ-secretase, generating the extracellular P3 peptide and releasing the APP intracellular domain (AICD) to the cytosol [2]. In the amyloidogenic pathway, APPs are cleaved by β-secretase (also called beta-site amyloid precursor protein cleaving enzyme (BACE)), which releases the soluble APPβ (sAPPβ) fragment into the extracellular space while leaving the rest of the protein (CTFβ) on the plasma membrane. Further cleavage of CTFβ by γ-secretase generates Aβ peptides, composed of 40–43 amino acids [6] (Figure 1). In the pathogenesis of AD, such as brain damage and cognitive impairment, Aβ peptides are abnormally aggregated and accumulated in the brain. Thus, these peptides in the brains of AD patients have been proposed as key targets for the treatment of AD.

For the past hundred years since AD was discovered, many global pharmaceutical companies have tried to develop drug candidates that can regulate abnormal Aβ aggregation. They have manufactured several synthetic compounds that inhibit the progression of Aβ-plaque formation, including LY2886721 [7], AN-1792 [8], and Verubecestat [9]. When these compounds were administered to patients with mild cognitive impairment (MCI), which is the pre-stage of AD, they have showed several good effects for pharmacological acceptability. However, obvious side effects, such as meningoencephalitis, neuronal hyperactivity, and liver toxicity, were observed in pre-clinical and clinical trials [7,8,9], and thus 99.6% of these products have been declared unfit for the treatment of AD. Recently, natural products have been probed to find safe and inexpensive therapeutics that are suitable for mass production and non-invasive treatment [10]. Development of such therapeutics using natural products is currently focused on the two different approaches—targeting the secretases related to Aβ aggregation or directly interfering with Aβ aggregates. In this review, we introduce several natural products, targeting various steps in each pathway, as potential therapeutics against AD.

## 2. Natural Products Targeting Secretases (Secretase-Dependent Approaches)

α-secretase cleaves APP at the transmembrane region. This region includes a disintegrin and metalloprotease (ADAM) domain, which are expressed on the surfaces of cells and anchored in the cell membrane. In most cases, α-secretase selectively targets ADAM10 [11]. As the cleavage of APPs by α-secretase results in the generation of the extracellular P3 peptide and intracellular AICD fragments without Aβ deposition in the non-amyloidogenic pathway, increasing the activity of α-secretase has recently been considered as a therapeutic approach for the treatment of AD.

On the other hand, β-secretase, also called beta-site amyloid precursor protein cleaving enzyme (BACE), is a major factor involved in the production of Aβ peptides in the amyloidogenic pathway [2]. There are two types of BACE—BACE1 and BACE2, which have comparable structures. In the cleavage pattern of APP, the level of CTFβ is more upregulated by the activation of BACE1 than the activation of BACE2 [12]. Several studies have reported that blocking of BACE1 ameliorates AD by suppressing Aβ production. In pre-clinical trials, BACE1-deficient mice did not exhibit cognitive impairment and accumulation of Aβ plaques [13]. Based on these results, many pharmaceutical companies have begun developing drug candidates that are derived from natural products with the preventive effect against AD by suppressing BACE1.

In general, α- and β-secretases are competitively activated during the process of APP cleavage in the brains of non-AD patients. The cleavage of APP by α-secretase precludes the production of Aβ in the brains of AD patients. Therefore, either increasing the α-secretase activity or decreasing the β-secretase activity is a very promising strategy to suppress the neuropathological changes associated with most of the symptoms [14,15]. In this section, we will introduce seven natural compounds that can regulate Aβ production in AD by modulating the activity of α- or β-secretase.

### 2.1. Ginsenoside Rg1

Ginsenosides are triterpene saponin (glycoside) components obtained from the root plant ginseng (*Panax ginseng*) and have been extensively used as a traditional medicine for thousands of years in East Asian countries. Several major functions of the ginsenosides contain pharmacological effects and improved immune systems including anti-aging, anti-stress, wound healing, antioxidant effects, and anti-inflammatory [16]. Ginsenosides have many species, but it is suggested that a few compounds, such as ginsenoside Rb1, Rb2, Rc, Rg1, and Rg3, are able to inhibit the acetylcholinesterase, which is one of the main neurotransmitters in the synaptic neurons of the brain, and butyrylcholinesterase, and therefore have potential for therapy of AD [17]. Ginsenoside Rg1 (Figure 2), especially proven as noticeable extracts compounds for the ingredients of AD treatments, mainly targets the activation of the transcription factor PPAR-γ through binding to the *BACE1* promoter instead of BACE1, eventually leading to the prevention of the formation of toxic Aβ aggregates. In addition, it is reported that ginsenoside Rg1 at the range of 0.25 to 25 μM increases the level of α-secretase while decreasing the level of β-secretase up to 38% in N2a/APP695 cells (N2a cells transfected with the human APP gene) by regulating the level of CTFα and CTFβ, respectively [18]. The level of Aβ in the cerebra of transgenic AD mice over-expressing APPs at age of 6 months (developing Aβ deposits in the brain) are significantly reduced concomitant with improvement in the cognitive deficit of the mice, after injecting them with ginsenoside Rg1 (10 mg/kg) for three months [18,19]. However, clinical trials with positive results have not been reported yet.

### 2.2. 2,2′,4′-Trihydroxychalcone (TDC)

TDC (Figure 3) is a natural product extracted from *Glycyrrhiza glabra* and belongs to the chalcone family of compounds. *Glycyrrhiza glabra* is common called licorice root in plant native of Asia and included pea family. This compound is known to have antiviral, anti-diabetic, antioxidant, and anti-inflammatory bioactivities [20]. Additionally, TDC has neuroprotective effects both in vitro and in vivo. In vitro, TDC decreases the levels of the BACE1 products sAPPβ and CTFβ in a concentration dependent manner, finally leading to the reduction of Aβ40 and Aβ42 levels. However, TDC has no effect on α- or γ-secretase activity. In vivo studies conducted administering TDC to two APP/PS1 mouse groups at a concentration of 9 mg/kg/day and 3 mg/kg/day. APP/PS1 AD mice, one of the AD mice models, and wild type mouse groups were administrated with vehicle. After 100 days injection, the cognitive abilities of all of mouse were progressed to the Morris Water Maze test. The results observed that memory impairment was alleviated when the dose was high. When higher dose administered, memory impairment was alleviated. Furthermore, no side effect, such as significant body loss, was observed in another animal trial with TDC administration, suggesting that TDC has no obvious toxicity in these animals [21]. However, little is known about the detail mechanism between BACE1 and TDC, and there is still a few of clinical trials targeting only AD’s with extracts of these ingredients.

### 2.3. Hispidin

Hispidin (Figure 4) is extracted from the fungus *Phellinus linteus*; it is a classified fungus. This natural compound is known as an antioxidant that eliminates toxic free radicals such as reactive oxygen and nitrogen species [22]. Additionally, it has been found to function as an inhibitor of BACE1 in a concentration-dependent manner without affecting the activities of α- and γ-secretases. This enzyme assay was conducted with a 50% inhibition concentration (IC_50_) is 4.0 × 10^−5^ through a kit from PanVera, WI, USA. Hispidin also improves learning and memory by suppressing vasopressin, substance P, and thyrotropin-releasing hormone, thereby affecting the levels of neuropeptides in the brain. However, hispidin is not small enough to pass through the blood–brain barrier (BBB) [23]. Therefore, further investigation for a small hispidin derivative with the same BACE1-inhibitory effect as that of hispidin is needed.

### 2.4. Berberine

Berberine (Figure 5) is an isoquinoline alkaloid extracted from *Coptidis rhizoma, Berberis vulgaris* L. [24,25]. This plant is included the flowering plant which is family of Ranunculaceae. It is generally used in Chinese herbal medicine for its anti-diarrheal, antimicrobial, and anti-inflammatory effects. In addition, Berberine is known to have a significant inhibitory activity against acetylcholinesterase. Acetylcholine is found at mainly neuromuscular junctions and in cholinergic-type chemical synapses, where it functions to terminate synaptic transmission. It is the primary target of inhibition by organophosphorus compounds, such as nerve agents, and acetylcholinesterase inhibitors are currently the available drugs used for the treatment of AD. Extracellular Aβ level was reduced by the treatment of berberine (IC_50_ = 5 μM and 50 μM) in cultured medium of APPNL-H4 cells, one of neuroglioma cell. Berberine treatment effectively reduced levels of Aβ_40_ (47.1 ± 11.5% at 5 μM; 35.5 ± 8.1% at 50 μM) and Aβ_42_ (49.1 ± 13.6% at 5 μM; 35.3 ± 10.2% at 50 μM); its 50% inhibition concentration (IC_50_) for extracellular Aβ production was around 5 μM [25]. When 25 mg/kg per day and 100 mg/kg per day for 4 months were administered by gavage in the AD transgenic mice at the age of 2 months, cognitive impairment was mitigated without increasing BACE1 levels [24]. Another animal trial was conducted with twenty-four male New Zealand white rabbits. A fresh suspension in saline solution of berberine chloride (50 mg/kg and 100 mg/kg) was administered intragastrically once daily for 14 days. In the brain of an aged AD rabbit model, berberine also reduces the BACE1 activity, thereby clearing the Aβ deposition, concurrent with improvement in the behavioral symptoms, such as head-tilting, tremor, weight loss, and paralysis [26,27]. Additional studies on animals have reported that berberine can help to manage type 2 diabetes and high blood cholesterol level, which are both contributing factors to accelerated aging and AD. Many research groups have suggested that berberine could be explored for use in AD patients. Clinical trials are reported to treat AD using berberine (Clinical trial ID. NCT03221894), but the results are not fully known, so more specific plans are needed.

### 2.5. Ligustilide

Ligustilide (Figure 6) is a secondary metabolite which is derived from the *Ligusticum chuanxiong Hort.,* belonging to *Umbelliferae* family of medicinal plants. It has been known to have an anti-inflammatory effect by inhibiting the secretion of TNF-α, which is one of the major proinflammatory cytokines, and TNF-α–induced nuclear factor (NF)-κB activation [28]. Several studies have demonstrated that ligustilide has a significant neuroprotective activity against brain damage in various stroke models and alleviates the cognitive deficit in a chronic cerebral hypoperfusion model [29,30]. Moreover, oral administration of ligustilide treatment with 10 or 40 mg/kg for 8 weeks ameliorates the neurotoxicity of Aβ in APP/PS1 mice, one of AD transgenic mice models [31]. Another research has suggested that ligustilide promotes the non-amyloidogenic processing of APP in vitro and in vivo by increasing the activity of α-secretase to cleave full-length APP. It ensures to target only ADAM10, without affecting the other proteins involved in APP cleavage. In another behavioral study with the APP/PS1 AD mouse model in comparison with wild type mice, oral administration of ligustilide with 30 mg/kg improved the neurobehavioral deficits. In addition, ligustilide treatment increased the ADAM10 levels and activity in APP/PS1 AD mice, consequently promoting the non-amyloidogenic pathway [32]. Relevant results of clinical trials have been rarely reported yet (Clinical trial ID. NCT01221662), but the neuroprotective effect of ligustilide through downregulation of Aβ aggregates and the consequent cognitive improvement strongly suggests that this natural compound is a potential therapeutic candidate against AD.

### 2.6. Epigallochetchin-3-Galate (EGCG)

Epigallocatechin-3-gallate (EGCG) (Figure 7) is a one of the most abundant compounds derived from green tea (*Camellia sinensis leaf*). The polyphenol structure of EGCG has been studied numerously for the treatment of neurodegenerative disorders including AD [33]. It has been proven to increase the secretion of α-secretase, thereby preventing the formation of abnormal Aβ aggregates [34]. Moreover, it prevents the amyloidogenic pathway by reducing the activity of β-secretase, which promotes abnormal aggregation of Aβ [35]. Pre-clinical trials using the APP/PS1 AD mouse model including oral administration of EGCG (50 mg/kg) have confirmed that it reduces the abnormal aggregation of Aβ in the brain and EGCG, which has been proven as the strong inhibitor of Aβ aggregation [36], is currently in phase 3 clinical trial. (Clinical trial ID. NCT00951834)

### 2.7. Polymethoxyflavones

Polymethoxyflavones (Figure 8), which have been widely used for the management of allergy or viral infections, are secondary metabolites extracted from the black ginger (*Kaempferia parviflora*) or citrus peels. Recently studies showed that they have more effects such as antioxidant, anti-fatigue, antiviral, anti-allergenic, and anti-mutagenic effects [37]. They consist of the three of major compounds: 5,7-dimethoxyflavone (DMF), 5,7,4′-trimethoxyflavone (TMF), and 3,5,7,3′,4′-pentamethoxyflavone (PMF). These three molecules have a common structure, which comprises two methoxy groups at C-5 and C-7 of the A ring in polymethoxyflavones that contributes to the inhibition of BACE1. In addition, the methoxy group at C-4′ on the B ring of 5,7,4′-trimethoxyflavone can particularly increase the inhibitory effect on BACE1. Any of these three compounds can bind to a particular regulatory site on BACE1, subsequently reducing the enzymatic function of this secretase in the amyloidogenic pathway [38]. Moreover, they do not affect the activities of other enzymes, such as α-secretase and serine protease, while inhibiting the BACE1-related amyloidogenesis. However, there are few clinical trials on the use of the polymethoxyflavones from *Kaempferia parviflora* against AD.

## 3. Natural Products Targeting Abnormal Aβ Aggregates (Structure-Dependent Approaches)

During AD progression, Aβ peptides aggregate into oligomers, protofibrils, and fibrils that are heterogeneous in size and structure. A common feature of all these aggregates is the formation of cross-β structures, which are extended β-sheets with two β-strands aligned in horizontal orientation to the fibril axis [39,40,41]. Each β-strand has intramolecular hydrophobic interactions between the hydrophobic core domain (residues 16-20, KLVFF) and the hydrophobic C-terminus of Aβ peptides, and intermolecular hydrogen bonding among the β-sheets [42,43,44]. Phenylalanine 19 (F19) and 20 (F20) residues in the hydrophobic core domain are essential for these interactions because these two residues are hydrophobic and aromatic [40,45]. The hydrophobicity is thought to be the primary driving force of abnormal Aβ aggregation, which might be supported by the aromaticity [46,47] (Figure 9). Thus, targeting F19 and F20 residues of Aβ may be important to change the structure of Aβ aggregates and reduce the associated toxicity.

In the following section, we will introduce five natural compounds whose phenyl rings target the phenylalanine residues in the hydrophobic core domain of Aβ. These compounds all form hydrophobic interactions or hydrogen bonds with Aβ, consequently destroying common structure of Aβ fibrils and preventing its toxicity on AD.

### 3.1. Resveratrol

Resveratrol (Figure 10) is a polyphenol stilbene abundantly found in many edible plants, such as grapes (*Vitis vinifera*) of the *Vitaceae* family and peanuts (*Arachis hypogaea*) of the *Arachis* family. It has neuroprotective as well as anti-inflammatory and antioxidant effects and thus may be useful in the treatment of AD [48,49]. Resveratrol decreases the formation of Aβ plaques [50,51,52] and lowers their toxicity in various ways, but the underlying mechanisms are still unclear. Although resveratrol initially accelerates the formation of Aβ fibrils with low toxicity via hydrophobic interaction between its phenolic ring and Aβ hydrophobic domain, these fibrils then undergo conformational changes into unstructured aggregates by the self-stacking of aromatic residues in Aβ core domain, consequently reducing the level of aggregation. Resveratrol is also known to suppress the phosphatidylinositol 3-kinase/protein kinase B signaling pathway in addition to having other effects [53,54] that are closely linked to AD, such as the reduction of the amount of Aβ aggregates or Aβ toxicity. In clinical trials for the treatment of AD, oral administration of low doses resveratrol (5 mg/day) was found safe and well tolerated, but less effective on ameliorating pathology of AD (Clinical trial ID. NCT00678431) [55]. Therefore, further research on the use of resveratrol as a therapeutic candidate against AD and on the signaling pathways involved in the anti-AD effect of resveratrol are needed.

### 3.2. Brazilin

Brazilin (Figure 11) is a major compound that can be obtained from the Sappan wood (*Caesalpinia sappan*). It has been used as a red dye in the Middle Ages but has recently been studied to prevent abnormal Aβ aggregation [56]. It is suggested that brazilin can inhibit the fibrillogenesis of Aβ and decrease the cytotoxicity. Moreover, brazilin has a role in remodeling the mature fibril of Aβ into non-functional structures. This structural change of Aβ is related to their interactions with chemical structures and residues of brazilin. Brazilin has four hydroxyl groups, which have a powerful hydrophobic interaction with the phenyl ring in F20 of Aβ. This interaction results in ≥10 hydrogen bonds and causes conformational changes in Aβ, consequently lowering the neuronal toxicity of Aβ. Through molecular dynamics simulations, it can be also verified that direct binding between brazilin and L17, F19, F20, and K28 residues in the hydrogen bonds of Aβ is observed in Aβ pentamer fibril structures [57]. Therefore, numerous studies aiming to improve the neuroprotective effects of brazilin and possibility of development as the drug of AD have been conducting but reports of clinical trials released positive results does not appear to have been registered yet.

### 3.3. Curcumin

Curcumin (Figure 12), one of the polyphenol compounds, is the primary curcuminoid compared with other secondary metabolites, demethoxycurcumin and bis-demethoxycurcymin, produced by turmeric (*Curcuma longa*) of the ginger family (*Zingiberaceae*). It is usually used against AD in the oriental traditional medicine [58]. Two feruloyl groups of curcumin structurally resemble phenylalanine [59] and thus interact with the hydrophobic core domains of Aβ oligomers [60,61]. Curcumin at the concentration of 1 μM hydrophobically interactions with F19 and F20 in the hydrophobic core domain of Aβ. At high concentrations (15 μM), it forms hydrogen bonds with the hydrophobic core domain of Aβ in addition to F19-F20 hydrophobic interaction [60]. The interaction of curcumin with Aβ oligomers does not suppress Aβ aggregation but disrupts their β-sheet structure, thereby alleviating their toxicity [61]. Curcumin shows bioactivity in vitro, such as anti-oxidant and anti-amyloid effects [60,61,62], and in Tg2576 APPSw transgenic mouse, one of the AD mouse models, fed curcumin (1–8 μM) [62,63]. However, oral administration of curcumin (each 2 g/day or 4 g/day) has no effect on the formation of Aβ aggregates in AD patients (Clinical trials ID. NCT00099710) [58]. This result may be due to the short half-life and low bioavailability of curcumin. Thus, further research about other curcumin preparations should be performed to use this natural compound as a therapeutic against AD.

### 3.4. Tannic Acid

Tannic acid (Figure 13) is generally found in the galls of *Quercus* species, for example, aleppo oak (*Quercus infectoria*) and golden oak (*Quercus alnifolia*) of the *Fagaceae* family. It accumulates in large amounts in specific tissues and organs, such as barks, fruits, leaves, or roots. It is a type of polyphenol with aromatic residues and is known to interact with Aβ fibrils, whereby the formation of these fibrils is inhibited [40,45]. Although the underlying mechanism is still unclear, it is certain that the aromatic residues of tannic acid undergo aromatic stacking with the hydrophobic domain of Aβ. Tannic acid may bind to the extending soluble Aβ oligomers as well as fibrils. It destabilizes the conformation of Aβ fibrils and promotes the depolymerization of Aβ into non-toxic or toxic species in vitro polymerization assay [64,65,66]. Even if toxic oligomers are formed, tannic acid can reduce Aβ toxicity via its anti-toxic bioactivity. Additionally, tannic acid can bind to Aβ monomers and consequently prevent their polymerization into fibrils [65]. Although, the effect of tannic acid in clinical trials with AD patients is not widely conducted yet, it mostly prevents the development of AD by reducing the levels of reactive oxygen species and abnormal Aβ aggregates via its antioxidant and anti-amyloidogenic activities in vivo [67].

### 3.5. Theaflavin

Theaflavin (Figure 14) is a group of polycyclic polyphenols extracted during the fermentation of tea into black tea. This polycyclic phenol has a higher bioavailability than other types of polyphenols extracted from other plants. Several studies have revealed that theaflavin has protective effects in neurodegenerative diseases, including AD and Parkinson’s disease, in addition to its antioxidant and antiapoptotic effects [68,69]. Theaflavin interferes with the formation of Aβ oligomers and fibrils early during AD development. It binds to the non-polar hydrophobic amino acids in the hydrophobic core domain and hydrophobic C-terminus of Aβ. These interactions are based on hydrophobic interaction rather than sequence specificity [69]. Consequently, theaflavin prevents the formation of toxic Aβ aggregates and induces the remodeling of misfolded Aβ aggregates into new types of Aβ structures, which are spherical or amorphous aggregates. These Aβ aggregates with new structural conformations establish self-interactions among their hydrophobic core domains, eventually inhibiting the formation of cross-β structures of Aβ in vitro. Moreover, 15 μM of theaflavin can reduce the toxicity of abnormal Aβ aggregates in neural cells and cause conformational changes in preformed Aβ fibrils into other Aβ structures without neural cytotoxicity in neuronal cell culture [51,70]. Additionally, theaflavin can maintain its anti-amyloidogenic activity even after oxidation through prolonged exposure to air [68]. Accordingly, several studies have hypothesized that theaflavin might be used as a potent therapeutic for the treatment of AD. However, to date, few clinical trials for the use of theaflavin against AD have been reported.

## 4. Discussion

Here, we compared the effects and structures of the natural products that can modulate either of the two APP processing pathways (secretase- or structure-dependent) on the formation of the Aβ aggregates in AD and summarized the states of the corresponding clinical trials (Table 1 and Table 2). Natural products are expected to be less hazardous than synthetic compounds because such products have existed naturally and been utilized in traditional medicine for a long time [71,72]. Given the importance of Aβ in AD, numerous studies are centered on the molecular mechanisms underlying AD and related interventions. Natural products are especially studied as drug candidates for the treatment of AD. Above all, reducing the amount of abnormal Aβ aggregates, a typical phenomenon in AD patients, is the major aim in using natural products against AD.

Several natural products targeting the secretase-dependent mechanisms in the formation of abnormal Aβ aggregates affect the stimulation of the ADAM family of proteins (especially α-secretase) or the inhibition of BACE1 (also called β-secretase) during both amyloidogenic and non-amyloidogenic pathways in the cleavage of APP, thereby modulating the formation of these aggregates. As α- and β-secretases are commonly activated competitively in the APP-cleavage process, this competition can control the Aβ accumulation and alleviate the cognitive impairment in AD. This competitive mechanism is skewed toward the activation of α-secretase over β-secretase by natural products such as ginsenoside Rg1, TDC, hispidin, berberine, ligustilide, EGCG, and polymethoxyflavones. The structure-dependent pathway involves the interaction of natural products such as resveratrol, brazilin, curcumin, tannic acid, and theaflavin with misfolded Aβ structures to induce conformational changes in these structures. The interaction of such a natural product with the hydrophobic core domain of Aβ disrupts the cross-β conformation, altering the Aβ structure and thereby lowering the Aβ-induced neuronal toxicity.

As the studies that prove the efficacy of natural products increase, these compounds are increasingly suggested as therapeutic candidates against AD [10,73]. However, there are several drawbacks that need to be overcome to use a substance from nature as a medicine. The molecular weight of such natural products should be as small as possible. Compounds with high molecular weight cannot pass through the BBB. Additionally, they must be small enough to be absorbed by the body and to be properly activated in the cerebrum. However, most natural substances, such as TDC, have high molecular weight [20]. Therefore, technical approaches to derive small natural products (≤400 dalton) that can cross the BBB need to be developed. In addition, natural products must be mass-produced through the gene recombination technology to be utilized as commercial and global medicines. Moreover, as natural products include numerous substances with various functions for therapeutic properties, it is necessary to separate the single leading compounds which are specific to AD. Further investigation is needed to isolate and identify a specific single compound in natural products targeting abnormal aggregation of Aβ.

Hyperphosphorylation of tau proteins is another hallmark of AD. The hyperphosphorylation of tau proteins generates paired helical and straight microtubule filaments (SF) [74,75], which consequently aggregate to form neurofibrillary tangles in neurodegenerative disorders, including AD.

Additionally, activation of astrocytes and microglia have been observed in the regions of Aβ plaques and neurofibrillary tangles in the brains of AD patients, concurrent with elevated inflammatory cytokine and chemokine levels, suggesting that neuroinflammation is also one of the major causes of AD pathogenesis [76,77]. Taken altogether, it is necessary to identify natural products that modulate both the Aβ and tau abnormalities as well as neuroinflammation in AD.

## Figures and Tables

**Figure 1 ijms-22-02341-f001:**
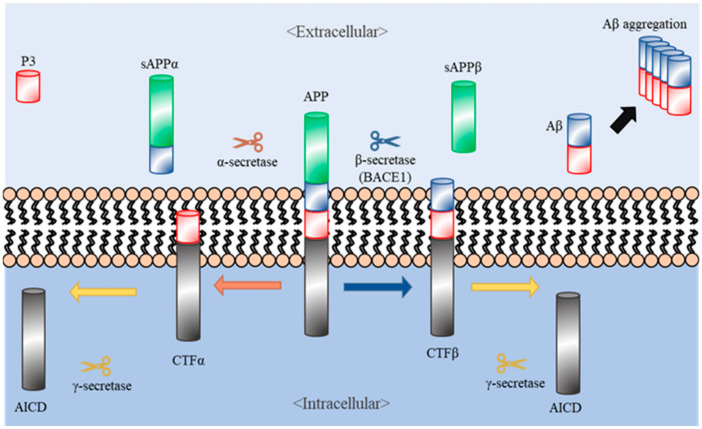
Amyloid precursor proteins (APP) and Alzheimer’s disease (AD). The left and right sides of the figure depict the non-amyloidogenic and amyloidogenic processing of APP (center), respectively. APP is cleaved by α- and γ-secretases, generating the products sAPPα, P3, and AICD (left side). Additionally, APP can be cleaved by β- (BACE1) and γ-secretases, generating the products sAPPβ, AICD, and Aβ (right side). AD symptoms occur because of excessive aggregation of Aβs (on the right).

**Figure 2 ijms-22-02341-f002:**
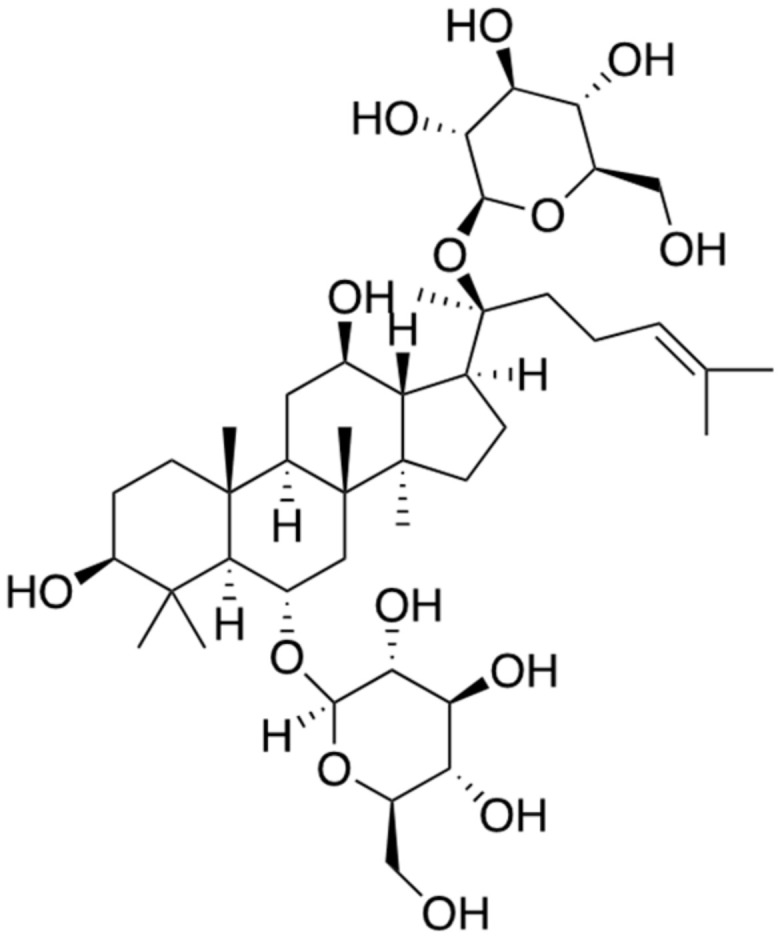
Chemical structure of ginsenoside Rg1.

**Figure 3 ijms-22-02341-f003:**
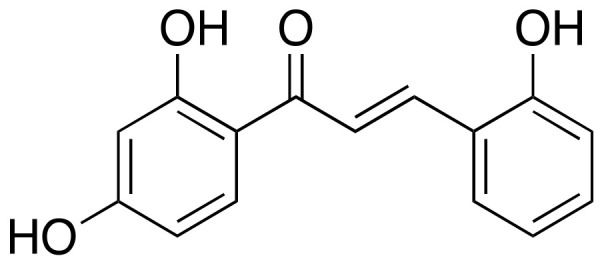
Chemical structure of 2,2′,4′-trihydroxychalcone (TDC).

**Figure 4 ijms-22-02341-f004:**
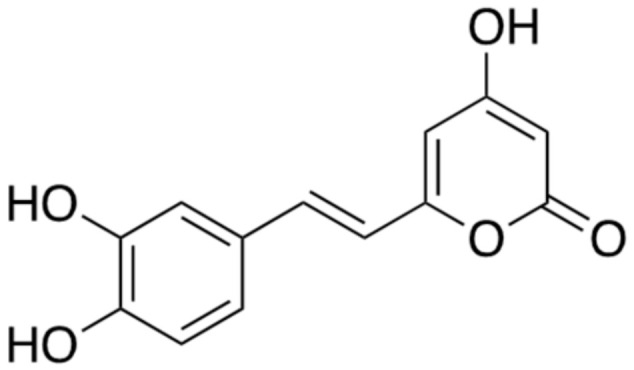
Chemical structure of hispidin.

**Figure 5 ijms-22-02341-f005:**
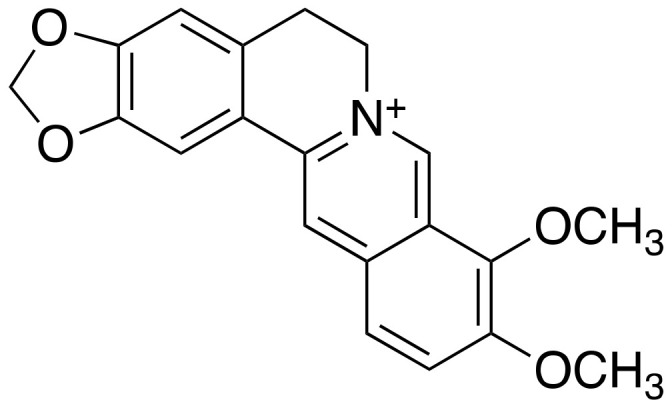
Chemical structure of berberine.

**Figure 6 ijms-22-02341-f006:**
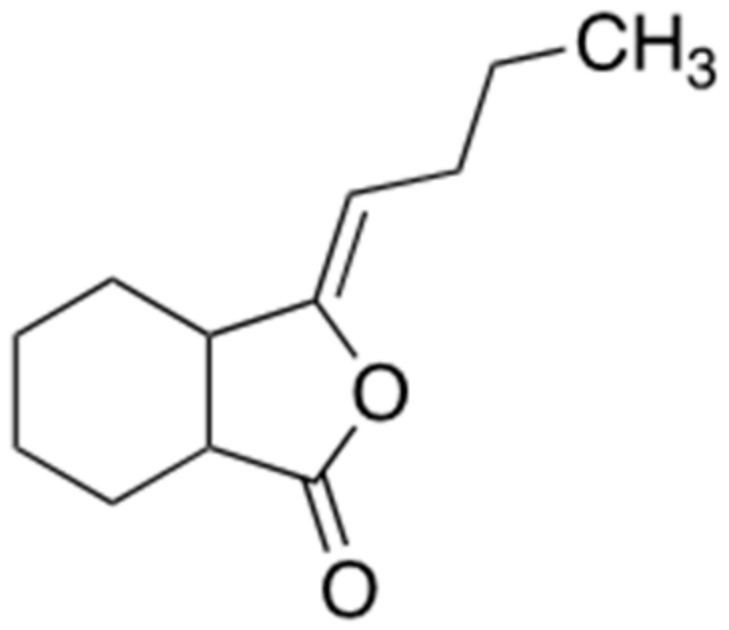
Chemical structure of ligustilide.

**Figure 7 ijms-22-02341-f007:**
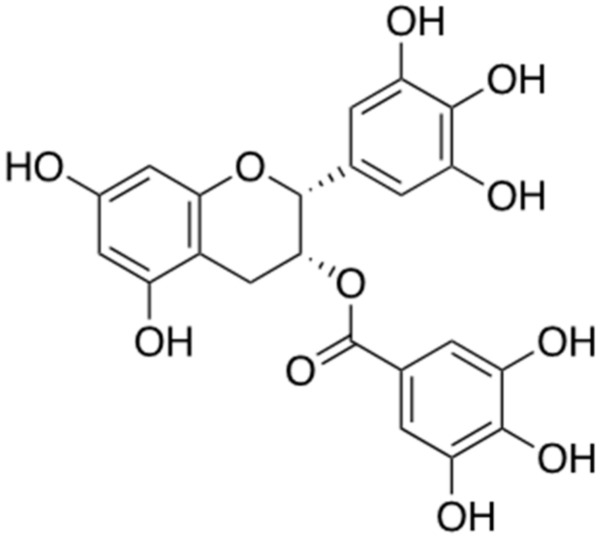
Chemical structure of epigallochetchin-3-galate (EGCG).

**Figure 8 ijms-22-02341-f008:**
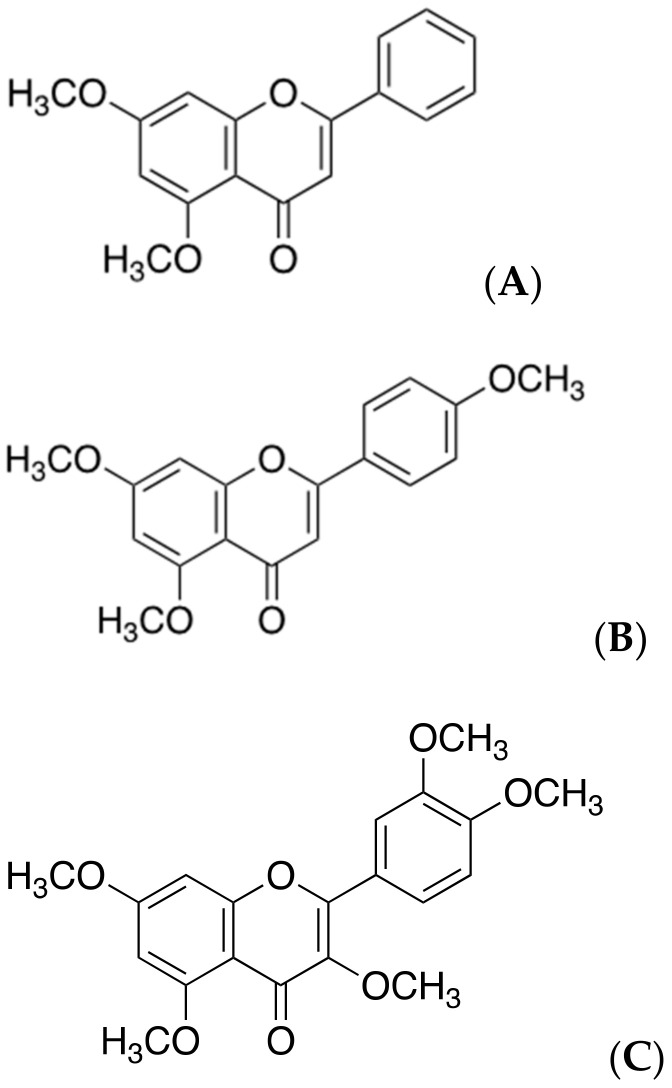
Chemical structures of polymethoxyflavones. Structures of (**A**) 5,7-methoxyflavone (DMF), (**B**) 5,7,4′-trimetholxyflavone (TMF), and (**C**) 3,5,7,3′,4′-pentamethoxyflavone (PMF).

**Figure 9 ijms-22-02341-f009:**
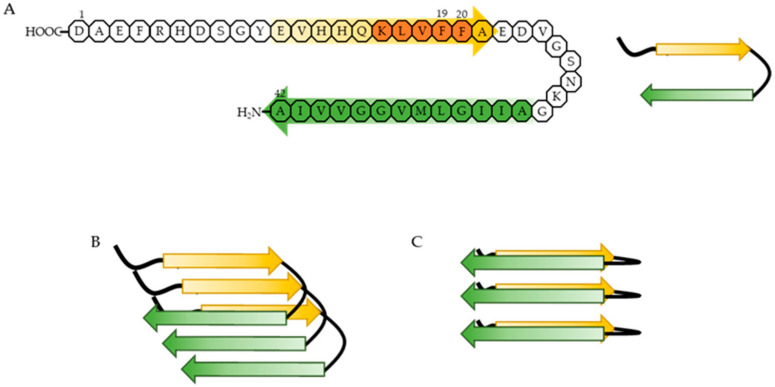
Proposed mechanism of abnormal Aβ aggregation. (**A**) Aβ1-42 forms a β-hairpin, including two β-strands. One of the strands is composed of residues 11–24 (yellow arrow) and contains the hydrophobic core domain (orange domain), and the other is composed of residues 30–42 (green arrow), regarded as the hydrophobic C-terminus. The hydrophobic core domain and the hydrophobic C-terminus have hydrophobic interactions. (**B**) Aβ in its β-hairpin conformation aggregates into soluble Aβ oligomers and forms β-sheets via intermolecular hydrogen bonding. (**C**) Soluble Aβ oligomers undergo a conformational change into an insoluble form with typical cross- β structure via intermolecular hydrogen bonding.

**Figure 10 ijms-22-02341-f010:**
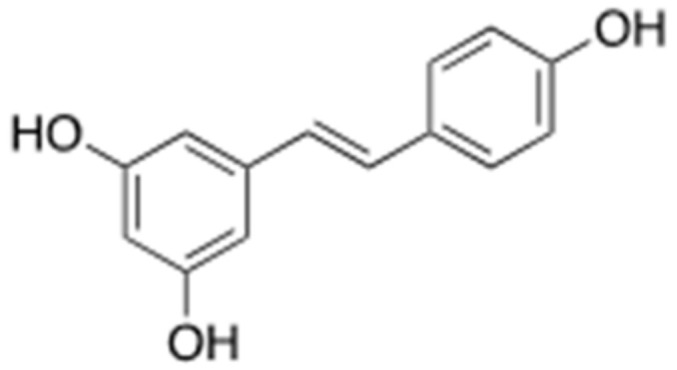
Chemical structure of resveratrol.

**Figure 11 ijms-22-02341-f011:**
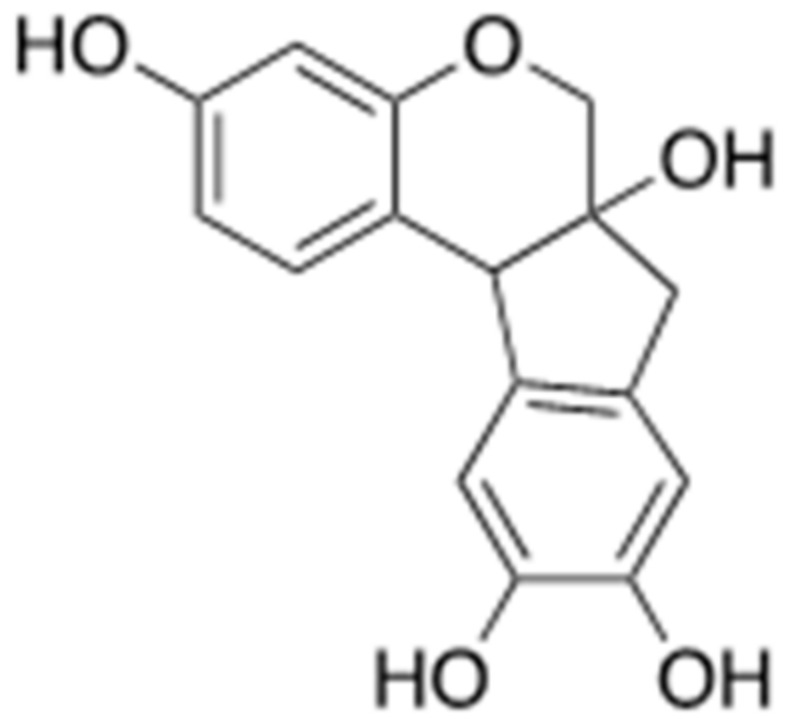
Chemical structure of brazilin.

**Figure 12 ijms-22-02341-f012:**
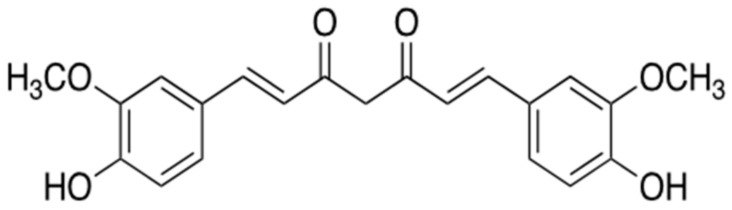
Chemical structure of curcumin.

**Figure 13 ijms-22-02341-f013:**
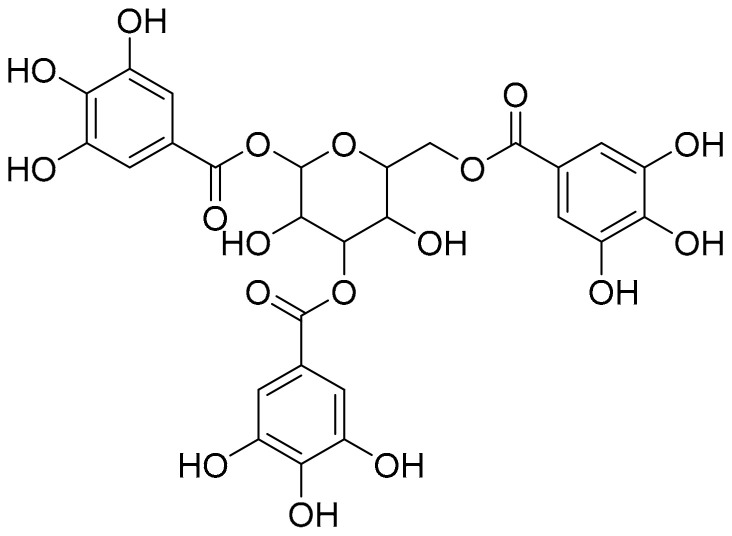
Chemical structure of tannic acid.

**Figure 14 ijms-22-02341-f014:**
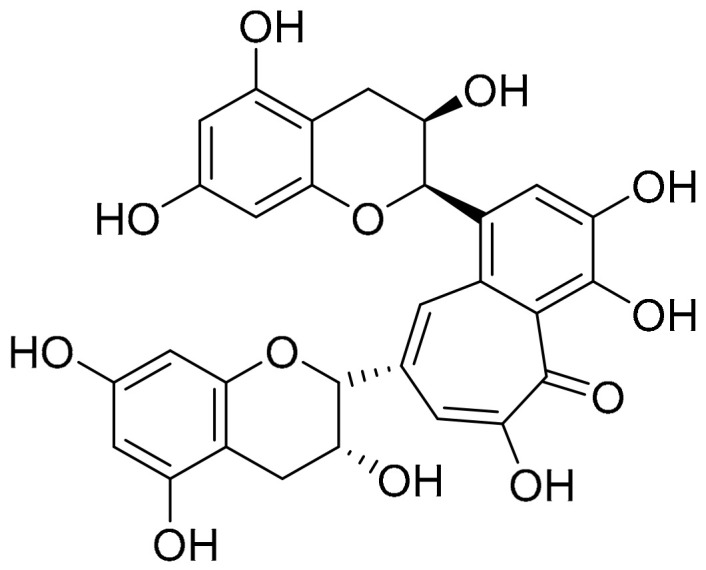
Chemical structure of theaflavin.

**Table 1 ijms-22-02341-t001:** The information of natural products targeting secretase-dependent ways in AD treatment.

Mechanism	Name (Origin)	ID	Concentration	Group	Time Span	Administration	Clinical Trial
Targeting to BACE1 or α-secretase (secretase-dependent ways) 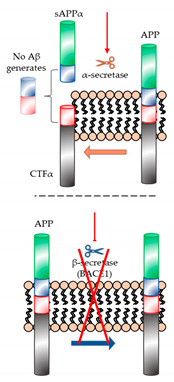	Ginsenoside Rg1 (*Panax ginseng*)	-	-	-	-	-	Improvement of level Aβ in the transgenic AD mice, but clinical trials with patients was not reported yet.
	2,2′,4′-Trihydroxychalcone (TDC) (*Glycyrrhiza glabra*)	-	-	-	-	-	Have no known clinical trials.
	Hispidin (*Phellinus linteus*)	-	-	-	-	-	Ongoing trial is not reported yet.
	Berberine (*Coptidis rhizoma*)	NCT03221894	Donepezil: 5~10 mg Berberine: 10 g/d	Donepezil group and Donepezil + grape granule group	12 months	Each bag of granules with 150 mL warm water melt was taken orally twice a day.	Currently little planned or ongoing clinical trials.
	Ligustilide (*Ligusticum chuanxiong Hort.**)*	-	-	-	-		Clinical trials have been rarely reported yet.
	Epigallochetchin-3-galate (EGCG) (*Camellia sinensis leaf*)	NCT00951834	EGCG: 200~800 mg/day (dependent on intake period)	Single group assignment (21 people)	18 months	Co-medication with donepezil	in phase 3
	Polymethoxyflavones (*Kaempferia parviflora*)	-	-	-	-	-	It is reported that there are not enough clinical trials using this.

**Table 2 ijms-22-02341-t002:** The information of natural products targeting structure-dependent ways in AD treatment.

Mechanism	Name (Origin)	ID	Concentration	Group	Time Span	Administration	Clinical Trial
Targeting to changes of structure. (structure-dependent way) 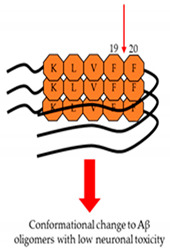	Resveratrol (*Arachis*)	NCT00678431	Resveratrol: 5 mg/day Dextrose and malate: 5 g/day	Placebo group and dextrose, malate, and resveratrol groups (32 people)	3, 6, and 12 months of each group	Administered twice a day in liquid form and taken in a 15ml volume dissolved in unsweetened grape juice	Its effects on AD are less deterioration.
	Brazilin (*Caesalpinia sappan*)	-	-	-	-	-	Was not reported yet.
	Curcumin (*Zingiberaceae*)	NCT00099710	Each 2 g/day or 4 g/day of curcumin	Placebo group and 2 g/day and 4 g/day groups (36 people)	24-weeks and 48-weeks of each group	Oral administration by capsules	Effects on Aβ aggregates was not observed in AD patients.
	Tannic acid (*Fagaceae*)	-	-	-	-	-	Clinical trials are not widely conducted yet.
	Theaflavin (the fermentation of tea into black tea)	-	-	-	-	-	The clinical trials for use of theaflavin as a drug candidate in AD treatment was little reported yet.

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
