# Peer review of "Natural Products Targeting Amyloid Beta in Alzheimer’s Disease"

_ijms, 2021, doi:10.3390/ijms22052341_

Round 1

Reviewer 1 Report

Dear Authors,

I have gone through your manuscript and I have an impression, that it does not give any complete set of knowledge. The manuscript should be carefully revised and many details should be added.

  1. Please, explain you selection of the natural products. What was the pattern for this selection. Add these information at the end of the chapter 2 and 3.
  2. how many years are covered by this review?
  3. which databases were screened for the relevant information?
  4. I recommend to transfer the structures of compounds on the top of every subchapter. it would be much easier to read about the influence of certain chemical groups on the activity, when seeing the structure next to the description
  5. please, add a little more introduction to each metabolite that you characterize: please mention the species that it comes from - both in English and in Latin and give the botanical family they belong to.
  6. it is very important to add to every compound the concentrations used in every described study.
  7. often it is not clear which animals were used for the tests, which administration route was selected,  or if the study was performed in vitro - in the cell cultures or in ready kits.
  8. it would be good to introduce better the characterized metabolites: which group of secondary metabolites they belong too, which structure are they based on, are they major components, in which polarity extracts they are present.
  9. concerning ginsenoside Rg1 - is it known if other saponins exhibit a similar potential? 
  10. the figures 3 and 4 do not bring any interesting information. add another column and point out the major mechanisms of the natural products' activity here in the figures, too. Add an additional column with the concentration used to achieve the described properties. Also add a column with references. the column 'clinical stage' does not bring any precious data...there are no clinical trials characterized here. add a column with the plant of origin and the botanical family.

other minor comments:

  • please, use italics for 'in vitro' and 'in vivo'
  • please, do not use capital letters for the names of secondary metabolites (e.g. 'berberine' instead of 'Berberine')

Author Response

Point-by-Point Responses

Reviewer’ commetns:

Reviewer #1 :

First of all, we would like to express our sincere gratitude for the time and effort which had put into reviewing our manuscript. Your insightful comments made this paper much more improved. All authors have carefully reviewed the comments in detail and revised the manuscript upon your meticulous advice.

Dear Authors,

I have gone through your manuscript and I have an impression, that it does not give any complete set of knowledge. The manuscript should be carefully revised and many details should be added.

  1. Please, explain you selection of the natural products. What was the pattern for this selection. Add these information at the end of the chapter 2 and 3.

We greatly appreciate in-depth comment. We divided two categories based on the mechanism of Aβ production and aggregation. Aβ can be produced by the activity of enzymes such as BACE1, and it can be aggregated into oligomers and fibrils with structural changes. Therefore, we focused these two points in AD targeting natural products. We corrected the manuscripts in the end of chapter 2 and 3 as your suggestion. Detailed corrections are below;

  1. a) In this section, we will introduce seven natural compounds that can regulate Aβ production in AD by modulating the activity of α- or β-secretase (line 99 of page 3).
  2. b) In the following section, we will introduce five natural compounds whose phenyl rings target the phenylalanine residues in the hydrophobic core domain of Aβ. These all compounds form hydrophobic interactions or hydrogen bonds with Aβ, consequently destroying common structure of Aβ fibrils and preventing its toxicity on AD (line 260 of page 7).

Thank you for your kindly suggestion again.

  1. how many years are covered by this review?

We apologize for our insufficient explanation. This review manuscript will help a lot in the drug development for the treatment of Alzheimer’s Disease using natural products. We mainly collected and included all information related with natural products in AD therapeutic with recent research original and review articles less then 5 years. And we expect that there are still a lot of research going on, and we look forward to the future of this review in hopes of further research.

  1. which databases were screened for the relevant information?

We sincerely apologize for our incomplete explanation. Because sources of natural products introduced in this manuscript could be easily obtained in nature and had been empirically used for AD treatment as well as other several pathologies, they are recognized as traditional medicines. As science develops, effector molecules of natural plants have been discovered and their effects such as anti-inflammatory, anti-oxidant, and anti-amyloidogenic are proved in many research articles. Additionally, we obtained the information of clinical trials from the clinical articles and National Library of Medicine (www.clinicaltrials.gov). We included all information in the references.

  1. I recommend to transfer the structures of compounds on the top of every subchapter. it would be much easier to read about the influence of certain chemical groups on the activity, when seeing the structure next to the description

We deeply appreciate that you have pointed out what could help us make our manuscript more refined. According to your advice, the visual materials will be more readable. So, we modified the contents included structures of compounds as your suggestion (page 4).

  1. please, add a little more introduction to each metabolite that you characterize: please mention the species that it comes from - both in English and in Latin and give the botanical family they belong to.

We deeply appreciate for your valuable comments. We added brief introduction about each metabolites as your suggestion. Moreover, we added the more information such as classification species, and more scientific name. Detailed corrections are below;

  1. a) Ginsenosides are triterpene saponin (glycoside) components obtained from the root plants ginseng (Panax ginseng) and have been extensively used as a traditional medicine for thousands of years in East Asian countries. (line 103 of page 3)
  2. b) TDC is a natural product extracted from Glycyrrhiza glabra and belongs to the chalcone family of compounds. (line 126 of page 4)
  3. c) Hispidin is extracted from the fungus Phellinus linteus, it is classified fungus. (line 146 of page 4)
  4. d) Berberine is an isoquinoline alkaloid extracted from Coptidis rhizoma, Berberis vulgaris L.. This plant is included the flowering plant which is family of Ranunculaceae. (line 160 of page 5)
  5. e) Ligustilide is secondary metabolites which is derived from the Ligusticum chuanxiong Hort., belonging to Umbelliferae family of medicinal plants. (line 191 of page 5)
  6. f) Epigallocatechin-3-gallate (EGCG) is a one of the most abundant compounds derived from green tea (Camellia sinensis leaf). (line 212 of page 6)
  7. g) Polymethoxyflavones, which have been widely used for the management of allergy or viral infections, are secondary metabolites extracted from the black ginger (Kaempferia parviflora) or citrus peels. (line 225 of page 6)
  8. h) Resveratrol is a polyphenol stilbene abundantly found in many edible plants, such as grapes (Vitis Vinifera) of the Vitaceae family and peanuts (Arachis hypogaea) of the Arachis (line 272 of page 8)
  9. i) Brazilin is a major compound that can be obtained from the Sappan wood (Caesalpinia sappan). (line 292 of page 8)
  10. j) Curcumin, one of the polyphenol compounds, is the primary curcuminoid compared with other secondary metabolites, demethoxycurcumin and bis-demethoxycurcymin, produced by turmeric (Curcuma longa) of the ginger family (Zingiberaceae). (line 310 of page 9)
  11. k) Tannic acid is generally found in the galls of Quercus species, for example aleppo oak (Quercus infectoria) and golden oak (Quercus alnifolia) of the Fagaceae (line 331 of page 9)

  1. it is very important to add to every compound the concentrations used in every described study.

We really apologize for our incomplete information of each compound. We carefully reviewed again to correct or add the missing information in the revised version. We listed all the concentrations of the compounds used in the experiments. Detailed corrections are below;

  1. a) In addition, it is reported that ginsenoside Rg1 at the range of 0.25 to 25μM increases the level of α-secretase while decreasing the level of β-secretase up to 38% in N2a/APP695 cells (N2a cells transfected with the human APP gene) by regulating the level of CTFα and CTFβ, respectively. (line 114 of page 3)
  2. b) The level of Aβ in the cerebra of transgenic AD mice over-expressing APPs at age of 6 months (developing Aβ deposits in the brain) are significantly reduced concomitant with improvement in the cognitive deficit of the mice, after injecting them with ginsenoside Rg1 (10mg/kg) for three months. (line 118 of page 3)
  3. c) In vivo studies conducted administering TDC to two APP/PS1 mouse groups at a concentration of 9mg/kg/day and 3mg/kg/day. (line 133 of page 4)
  4. d) Related with hispidin enzyme assay was conducted with a 50% inhibition concentration (IC50) is 4.0×10-5 through a kit from PanVera, WI, USA. (line 150 of page 4)
  5. e) When 25mg/kg per day and 100mg/kg per day for 4 months were administered by gavage in the AD transgenic mice at the age of 2 months, cognitive impairment was mitigated without increasing BACE1 levels. Another animal trial was conducted with twenty-four male New Zealand white rabbits. A fresh suspension in saline solution of berberine chloride (50mg/kg and 100mg/kg) was administered intragastrically once daily for 14 days. (line 168 of page 5)
  6. f) Moreover, oral administration of ligustilide treatment with 10 or 40 mg/kg for 8 weeks ameliorates the neurotoxicity of Aβ in APP/PS1 mice, one of AD transgenic mice models. Another research has suggested that ligustilide promotes the non-amyloidogenic processing of APP in vitro and in vivo by increasing the activity of α-secretase to cleave full-length APP. It ensures to target only ADAM10, without affecting the other proteins involved in APP cleavage. In another behavioral study with the APP/PS1 AD mouse model in comparison with wild-type mice, oral administration of ligustilide with 30 mg/kg improved the neurobehavioral deficits. (line 195 of page 5)
  7. g) Pre-clinical trials using the APP/PS1 AD mouse model including oral administration of EGCG (50mg/kg) have confirmed that it reduces the abnormal aggregation of Aβ in the brain and EGCG, which has been proven as the strong inhibitor of Aβ aggregation, is currently in phase 3 clinical trial (Clinical trial ID. NCT00951834). (line 218 of page 6)
  8. h) In clinical trials for the treatment of AD, oral administration of low doses resveratrol (5mg/day) was found safe and well-tolerated, but less effective on ameliorating pathology of AD. (line 284 of page 8)
  9. i) Curcumin at the concentration of 1μM hydrophobically interactions with F19 and F20 in the hydrophobic core domain of Aβ. At high concentrations (15μM), it forms hydrogen bonds with the hydrophobic core domain of Aβ in addition to F19-F20 hydrophobic interaction. (line 315 of page 9) ~ Curcumin shows bioactivity in vitro, such as anti-oxidant and anti-amyloid effects, and in Tg2576 APPSw transgenic mouse, one of the AD mouse models, fed curcumin (1 ~ 8μM). However, oral administration of curcumin (each 2g/day or 4g/day) has no effect on the formation of Aβ aggregates in AD patients. (line 320 of page 9)
  10. j) Moreover, 15μM of theaflavin can reduce the toxicity of abnormal Aβ aggregates in neural cells and cause conformational changes in pre-formed Aβ fibrils into other Aβ structures without neural cytotoxicity in neuronal cell culture. (line 362 of page 10)

  1. often it is not clear which animals were used for the tests, which administration route was selected,  or if the study was performed in vitro - in the cell cultures or in ready kits.

We apologize for lacking details. We reflected on your comments and tried to add detailed experimental information in addition to the concentration of compounds used. Detailed corrections are below;

  1. a) The level of Aβ in the cerebra of transgenic AD mice over-expressing APPs at age of 6 months (developing Aβ deposits in the brain) are significantly reduced concomitant with improvement in the cognitive deficit of the mice, after injecting them with ginsenoside Rg1 (10mg/kg) for three months. (line 133 of page 4)
  2. b) In vivo studies conducted administering TDC to two APP/PS1 mouse groups at a concentration of 9mg/kg/day and 3mg/kg/day. (line 133 of page 4)
  3. c) Related with hispidin enzyme assay was conducted with a 50% inhibition concentration (IC50) is 4.0×10-5 through a kit from PanVera, WI, USA. (line 150 of page 4)
  4. d) When 25mg/kg per day and 100mg/kg per day for 4 months were administered by gavage in the AD transgenic mice at the age of 2 months, cognitive impairment was mitigated without increasing BACE1 levels. Another animal trial was conducted with twenty-four male New Zealand white rabbits. A fresh suspension in saline solution of berberine chloride (50mg/kg and 100mg/kg) was administered intragastrically once daily for 14 days. (line 173 of page 5)
  5. e) Moreover, oral administration of ligustilide treatment with 10 or 40 mg/kg for 8 weeks ameliorates the neurotoxicity of Aβ in APP/PS1 mice, one of AD transgenic mice models. Another research has suggested that ligustilide promotes the non-amyloidogenic processing of APP in vitro and in vivo by increasing the activity of α-secretase to cleave full-length APP. It ensures to target only ADAM10, without affecting the other proteins involved in APP cleavage. In another behavioral study with the APP/PS1 AD mouse model in comparison with wild-type mice, oral administration of ligustilide with 30 mg/kg improved the neurobehavioral deficits. (line 195 of page 5)
  6. f) Pre-clinical trials using the APP/PS1 AD mouse model including oral administration of EGCG (50mg/kg) have confirmed that it reduces the abnormal aggregation of Aβ in the brain and EGCG, which has been proven as the strong inhibitor of Aβ aggregation, is currently in phase 3 clinical trial (Clinical trial ID. NCT00951834). (line 218 of page 6)
  7. g) In clinical trials for the treatment of AD, oral administration of low doses resveratrol (5mg/day) was found safe and well-tolerated, but less effective on ameliorating pathology of AD. (line 284 of page 8)
  8. h) Through molecular dynamics simulations, it can be also verified that direct binding between brazilin and L17, F19, F20, and K28 residues in the hydrogen bonds of Aβ is observed in Aβ pentamer fibril structures. (line 301 of page 9)
  9. i) Curcumin shows bioactivity in vitro, such as anti-oxidant and anti-amyloid effects, and in Tg2576 APPS transgenic mouse, one of the AD mouse models, fed curcumin (1 ~ 8μM). However, oral administration of curcumin (each 2g/day or 4g/day) has no effect on the formation of Aβ aggregates in AD patients. (line 320 of page 9)
  10. j) It destabilizes the conformation of Aβ fibrils and promotes the depolymerization of Aβ into non-toxic or toxic species in vitro polymerization assay. (line 338 of page 10)

  1. it would be good to introduce better the characterized metabolites: which group of secondary metabolites they belong too, which structure are they based on, are they major components, in which polarity extracts they are present.

We deeply appreciate that you have pointed out what could help us make our manuscript more refined and improved. We went through additional verification of what you pointed out and made some corrections. Detailed corrections are below;

  1. a) Ligustilide is secondary metabolites which is derived from the Ligusticum chuanxiong Hort., belonging to Umbelliferae family of medicinal plants. (line 189 of page 5)
  2. b) Polymethoxyflavones, which have been widely used for the management of allergy or viral infections, are secondary metabolites extracted from the black ginger (Kaempferia parviflora) or citrus peels. (line 225 of page 6)
  3. c) Curcumin, one of the polyphenol compounds, is the primary curcuminoid compared with other secondary metabolites, demethoxycurcumin and bis-demethoxycurcymin, produced by turmeric (Curcuma longa) of the ginger family (Zingiberaceae). (line 310 of page 9)

  1. concerning ginsenoside Rg1 - is it known if other saponins exhibit a similar potential? 

We really apologize for the mistake which causing confusion. We also agree with you, so we have made some changes to reduce confusion in revised manuscript. It is known that saponins in ginseng is usually called ‘ginsenoside’ and ginsenoside include several kinds of compounds such as ginsenoside Ra and ginsenoside Rb. They have also many effects for improvement of health, so we added that relevant information. But we want to especially focus on the compound targeting the amyloid beta for the AD treatment. Thus, ginsenoside Rg1, which is the well-known as the agent of AD treatment, are covered by this manuscript.

We added this point in the manuscript and corrected into “Ginsenosides are triterpene saponin (glycoside) components obtained from the root plants Panax ginseng and have been extensively used as a traditional medicine for thousands of years ~ However, clinical trials with positive results have not been reported yet.” (line 103 of page 3)

  1. the figures 3 and 4 do not bring any interesting information. add another column and point out the major mechanisms of the natural products' activity here in the figures, too. Add an additional column with the concentration used to achieve the described properties. Also add a column with references. the column 'clinical stage' does not bring any precious data...there are no clinical trials characterized here. add a column with the plant of origin and the botanical family.

We sincerely apologize for inaccurate expression on figure. We revised the overall part of the figure. The compound structures in figure 3 and 4 were moved the subchapter, each of which was described, and the table added a summary of the experience or information on clinical trials. In addition, figures 3 and 4 were combined to enhance readability.

other minor comments:

  • please, use italics for 'in vitro' and 'in vivo'
  • please, do not use capital letters for the names of secondary metabolites (e.g., 'berberine' instead of 'Berberine')

Thank you for pointing out our mistake. We carefully reviewed and corrected the manuscript as your suggestion. Detailed corrections are below;

  1. a) Additionally, TDC has neuroprotective effects both in vitro and in vivo. (line 129 of page 4)
  2. b) Another research has suggested that ligustilide promotes the non-amyloidogenic processing of APP in vitro and in vivo by increasing the activity of α-secretase to cleave full-length APP. (line 197 of page 5)
  3. c) Extracellular Aβ level was reduced by the treatment of berberine (IC50=5 μM and 50 μM) in cultured medium of APPNL-H4 cells, one of neuroglioma cell. (line 168 of page 5)
  4. d) It is suggested that brazilin can inhibit the fibrillogenesis of Aβ and decrease the cytotoxicity. (line 294 of page 8)

Reviewer 2 Report

The paper offers a review of the current knowledge on the use of natural compounds to treat AD acting either on the activity of secretases or directly on the abnormal Aß aggregates.

The compounds are discussed individually and clustered according to their specific activity.

At the end of each paragraph, a brief description of the current state of in vitro and in vivo investigation is given, although active clinical trials (where there are any) should be clearly identified (providing the clinical trial ID number).

Often natural products contain not only one main active component, but also a complex mix of other substances that may contribute to the overall activity. Would the authors comment on this?

Overall the paper is well written, although there are some sentences that need rephrasing while only a few minor checks are necessary in other cases.

Line 12: "In particular, a number....." consider rephrasing to something like "In particular, a number of studies on natural products associated with AD, aiming to downregulate  Aß peptides and suppress the formation of abnormal Aß aggregates, thus reducing neural cell death, are being conducted.

Line 18 : remove they

Line 30: which are toxic to neural cells

Line 38: they then traslocate

Line 66: "they have shown the significant..." consider rephrasing

Line 70: I do not think as alternative medicines is required.

Line 73: either of

Line 79: correct durfaces to surfaces

Line 105: Ginsenoside mainly targets the activation of the transcription factor

Line 120: belongs

Line 145: not sure catalyses on its own makes much sense. Catalyses what?

Line 170: of

Line 177: through downregulation of

Line 178: suggests

Line 191: add citation

Line 200: add , after polymethoxyflavones. remove "and" and substitute with "that". I think the sentence is more clear that way.

Line 213: that they all involve add "the" before formation

Line 224-225: "targets" "forms" "hydrogen bonds"

Line 227: Proposed mechanism of abnormal Ab aggregation instead of hypothetical schematic of abnormal Ab aggregation

Line 231: (B)Ab in its beta-hairpin conformation aggregates into...

Line 233: into an insoluble form with typical cross-beta structure via intermolecular hydrogen bonding

Line 239: aggregation formation

Line 241: remodels

Lines 243-246: I am not sure I clearly understand the mechanism

Line 249: such as the reduction of the amount

Line 251: "but its effects..." not very clear

Line 263: with should be put at the end of the sentence

Line 264: hydrogen bonds

Line 276: interactions

Line 279: has been observed to have showing bioactivity in vitro, such as..., and being studied in vivo in AD animal models...

Line 297: has varied varies

Line 321: few clinical trials...have been reported

Line 340: intervening targeting

Line 358-359-361: molecular weight

Author Response

Point-by-Point Responses

Reviewer’ commetns:

Reviewer #2 :

The paper offers a review of the current knowledge on the use of natural compounds to treat AD acting either on the activity of secretases or directly on the abnormal Aß aggregates.

The compounds are discussed individually and clustered according to their specific activity.

First of all, we would like to express our sincere gratitude for your insightful comments that made this paper even better. All authors have carefully reviewed the comments in detail and revised the manuscript upon your meticulous advice.

At the end of each paragraph, a brief description of the current state of in vitro and in vivo investigation is given, although active clinical trials (where there are any) should be clearly identified (providing the clinical trial ID number).

We apologize for the mistake of not providing sufficient information. After a detailed investigation of the clinical trials, we gave references and additional explanations to materials that would be valuable for future research.

Often natural products contain not only one main active component, but also a complex mix of other substances that may contribute to the overall activity. Would the authors comment on this?

We really appreciate for your suggestion to improve our manuscript that we did not expected. As you said, most natural products include numbers of substances with various functions. They work together and are consequently effective on different biological pathologies as well as AD. However, it is necessary to study and isolate the only single compound leading the stronger effects on the treatment of AD in natural products. Thus, we introduce one main active component of natural products in this manuscript.

We added this point in the manuscript and corrected into “Moreover, as natural products include numerous substances with various functions for therapeutic properties, it is necessary to separate the single leading compounds which are specific to AD. Further investigation is needed to isolate and identify a specific single compound in natural products targeting abnormal aggregation of Aβ.” (line 414 of page 14)

Overall the paper is well written, although there are some sentences that need rephrasing while only a few minor checks are necessary in other cases.

We deeply apologize for all the mistakes in our manuscript. We also appreciate for reviewing details of manuscript despite your busy schedules. The quality of manuscript is improved thanks to your suggestions. We carefully reviewed and corrected the manuscript as your suggestion. The details corrections are below.

Line 12: "In particular, a number....." consider rephrasing to something like "In particular, a number of studies on natural products associated with AD, aiming to downregulate  Aß peptides and suppress the formation of abnormal Aß aggregates, thus reducing neural cell death, are being conducted.

We corrected as your suggestion.

Line 18 : remove they

We corrected as your suggestion.

Line 30: which are toxic to neural cells

We corrected as your suggestion.

Line 38: they then translocate

We corrected as your suggestion.

Line 66: "they have shown the significant..." consider rephrasing

We corrected as your suggestion.

they have shown the significant effects for pharmacological acceptability.

  • they have showed several good effects for pharmacological acceptability. (line 66 of page 22)

Line 70: I do not think as alternative medicines is required.

We deleted this part because it may provide misleading information for the reader.

Line 73: either of

We corrected as your suggestion.

Line 79: correct durfaces to surfaces

We corrected as your suggestion.

Line 105: Ginsenoside mainly targets the activation of the transcription factor

We corrected as your suggestion.

Line 120: belongs

We corrected as your suggestion.

Line 145: not sure catalyses on its own makes much sense. Catalyses what?

We apologize for our mistake that could provide misleading information for the readers by not being careful with the description. We understand your mention is about appropriate grammer.

“which catalyzes one of the main neurotransmitters, acetylcholine, in the synaptic neurons of the brain.”

  • “Inhibits one of the main neurotransmitters, acetylcholinesterase, in the synaptic neurons of the brain.” (line 109 of page 3)

Line 170: of

We corrected as your suggestion.

Line 177: through downregulation of

We corrected as your suggestion.

Line 178: suggests

We corrected as your suggestion.

Line 191: add citation

We apologize for our mistake that could provide misleading information for the readers by not being careful with the description. We added the information with clinical trial number and wish for enough information with your asked.

Line 200: add , after polymethoxyflavones. remove "and" and substitute with "that". I think the sentence is more clear that way.

We corrected as your suggestion.

Line 213: that they all involve add "the" before formation

We corrected as your suggestion.

Line 224-225: "targets" "forms" "hydrogen bonds"

We corrected as your suggestion.

Line 227: Proposed mechanism of abnormal Ab aggregation instead of hypothetical schematic of abnormal Ab aggregation

We corrected as your suggestion.

Line 231: (B)Ab in its beta-hairpin conformation aggregates into...

We corrected as your suggestion.

Line 233: into an insoluble form with typical cross-beta structure via intermolecular hydrogen bonding

We corrected as your suggestion.

Line 239: aggregation formation

We corrected as your suggestion.

Line 241: remodels

We corrected as your suggestion.

Lines 243-246: I am not sure I clearly understand the mechanism

We really apologize for our contents that could provide misleading information for the readers by not being careful with the description.

Once the central hydrophobic region of resveratrol accelerates the formation of fibrils, stacking of the aromatic residues in the Aβ hydrophobic core domain is induced. The interaction between Aβ and the phenolic ring of resveratrol may induce conformational changes in Aβ, consequently reducing the aggregation

  • “Although resveratrol initially accelerates the formation of Aβ fibrils with low toxicity via hydrophobic interaction between its phenolic ring and Aβ hydrophobic domain, these fibrils then undergo conformational changes into unstructured aggregates by the self-stacking of aromatic residues in Aβ core domain, consequently reducing the level of aggregation.” (line 277 of page 8)

Line 249: such as the reduction of the amount

We corrected as your suggestion.

Line 251: "but its effects..." not very clear

We corrected as your suggestion.

In clinical trials for the treatment of AD, low doses of oral resveratrol were found safe and well-tolerated, but its effects on AD are less deterioration and scarce.

  • Rephrased, “In clinical trials for the treatment of AD, low doses of oral resveratrol were found safe and well-tolerated, but less effective on ameliorating pathology of AD.” (line 286 of page 8)

Line 263: with should be put at the end of the sentence

We corrected as your suggestion.

Brazilin has four hydroxyl groups, with which the phenyl ring in F20 of Aβ can have a powerful hydrophobic interaction.

  • “Brazilin has four hydroxyl groups, which have a powerful hydrophobic interaction with the phenyl ring in F20 of Aβ.” (line 298 of page 9)

Line 264: hydrogen bonds

We corrected as your suggestion.

Line 276: interactions

We corrected as your suggestion.

Line 279: has been observed to have showing bioactivity in vitro, such as..., and being studied in vivo in AD animal models...

We corrected as your suggestion.

Line 297: has varied varies

We corrected as your suggestion.

Line 321: few clinical trials...have been reported

We corrected as your suggestion.

Line 340: intervening targeting

We corrected as your suggestion.

Line 358-359-361: molecular weight

We corrected as your suggestion.

Round 2

Reviewer 1 Report

Dear Authors,

thank you for the big work made on this manuscript.

I believe, that it brings more important data in its present form.

My last suggestion is to focus again on the table 2. Please, think again about its last shape. it does not bring any information, as majority of compounds are not clinically tested. Maybe you are able to add some more information to those lines where there are any clinical trials ongoing, e.g. group of patients, time span, the way of administration or other data. it would give  the table some more data...

Author Response

Point-by-Point Responses

Reviewer’s comments

#Reviewer 1

First of all, we greatly appreciate your deep interest in our manuscript and helpful suggestion. Your insightful suggestion made this manuscript much more improved. All authors have carefully reviewed the comments in detail and revised the manuscript upon your meticulous advice.

Dear Authors,

thank you for the big work made on this manuscript.

I believe, that it brings more important data in its present form.

My last suggestion is to focus again on the table 2. Please, think again about its last shape. it does not bring any information, as majority of compounds are not clinically tested. Maybe you are able to add some more information to those lines where there are any clinical trials ongoing, e.g. group of patients, time span, the way of administration or other data. it would give the table some more data...

We genuinely appreciate your valuable comments. We modified the table 1 and 2. Regarding your suggestion, we added the more information about clinical trial, and it will be beneficial to the reader. We wanted to find more clinical information, but not so much information existed because it was intended to lead to clinical trials based on previous research. However, we tried to contain as much information as possible, and we added all information which have been reported so far. Thank you again for your kindly comments.